# Spin environment of a superconducting qubit in high magnetic fields

S. Günzler [1,2,6] ✉, J. Beck [1,6], D. Rieger [1], N. Gosling [2], N. Zapata [2], M. Field [2], S. Geisert [2], A. Bacher [3,4], J. K. Hohmann [3,4], M. Spiecker [1,2], W. Wernsdorfer [1,2] & I. M. Pop [1,2,5] ✉

Superconducting qubits equipped with quantum non-demolition readout and active feedback can be used as information engines to probe and manipulate microscopic degrees of freedom, whether intentionally designed or naturally occurring in their environment. In the case of spin systems, the required magnetic field bias presents a challenge for superconductors and Josephson junctions. Here we demonstrate a granular aluminum nanojunction fluxonium qubit (gralmonium) with spectrum and coherence resilient to fields beyond one Tesla. Sweeping the field reveals a paramagnetic spin-1/2 ensemble, which is the dominant gralmonium loss mechanism when the electron spin resonance matches the qubit. We also observe a suppression of MHz range fast flux noise in magnetic field, suggesting the freezing of surface spins. Using an active state stabilization sequence, the qubit hyperpolarizes long-lived two-level systems (TLSs) in its environment, previously speculated to be spins. Surprisingly, the coupling to these TLSs is unaffected by magnetic fields, leaving the question of their origin open. The robust operation of gralmoniums in Tesla fields offers new opportunities to explore unresolved questions in spin environment dynamics and facilitates hybrid architectures linking superconducting qubits with spin systems.

Superconducting qubits have rapidly evolved from proof-of-concept demonstrations to precision-engineered devices within the cQED framework[1], featuring quantum non-demolition readout and real-time feedback. These advances have enabled the observation of quantum jumps and trajectories[2-4], active feedback error correction[5-7] and the exploration of quantum mechanics foundations[8-10]. Such precise control renders superconducting circuits ideal for interfacing with other mesoscopic degrees of freedom (DOFs), which may be deliberately integrated into hybrid architectures or arise from spurious microscopic systems that impair qubit performance. Hybrid quantum architectures, where superconducting circuits couple to less amenable but longer-lived, magnetic-field-sensitive DOFs, have already demonstrated impressive achievements, such as coherent spin-photon interactions[11-13], spin ensemble[14-16] and even single-spin detection[17,18] using superconducting resonators, as well as single-magnon detection with a superconducting qubit[19]. Concurrently, various spurious environmental DOFs with often unknown magnetic field susceptibility are pervasive in superconducting devices. These include quasiparticles[20-23], charge offsets[20,24], spins[25-31] and other TLS environments[32-35].

High magnetic fields offer a powerful tool to characterize and tune various DOFs coupled to superconducting qubits, yet they are rarely utilized. This is explained by the fragility of aluminum-based devices in magnetic fields, as the superconducting gap is suppressed at ~10 mT in bulk, and the Josephson junction (JJ) critical current diminishes in a Fraunhofer pattern. Utilizing thin aluminum films can improve field compatibility[36-38], nevertheless, it still entails significant

[1]PHI, Karlsruhe Institute of Technology, Karlsruhe, Germany. [2]IQMT, Karlsruhe Institute of Technology, Karlsruhe, Germany. [3]IMT, Karlsruhe Institute of Technology, Karlsruhe, Germany. [4]KNMFi, Karlsruhe Institute of Technology, Karlsruhe, Germany. [5]Physics Institute 1, Stuttgart University, Stuttgart, Germany. [6]These authors contributed equally: S. Günzler, J. Beck. ✉e-mail: simon.guenzler@kit.edu; ioan.pop@kit.edu

suppression of the qubit spectrum and coherence in the range of few hundred mT. While the reduction of the superconducting gap can be mitigated by using field-resilient, low-loss superconductors like Nb[18], granular aluminum (grAl)[39] or NbTiN[40,41], finding a source of non-linearity that maintains resilience under magnetic fields is considerably more challenging. Efforts to develop field-resilient JJs that avoid Fraunhofer interference patterns include gate-tunable JJs based on semiconducting nanowires[42,43] or graphene layers[44]. However, these JJs have shown marginal coherence, with qubit spectra exhibiting significant instability.

We overcome these limitations by using a grAl nanojunction fluxonium qubit, known as gralmonium. This qubit combines the grAl field resilience[45] with the unique benefits of the grAl nanojunction[46]: low microwave losses and a compact nanoscopic footprint that eliminates Fraunhofer interference. We measure energy decay times $T_1 \approx 8\,\mu s$ and coherence times $T_{2E} \approx 3\,\mu s$, robust in fields beyond 1 T, with less than 2% qubit frequency shift in this entire range. We identify a paramagnetic spin ensemble coupled to the gralmonium, showcasing its potential for sensing. We also observe a decrease in the fast flux noise in Hahn echo experiments in magnetic field, indicating a freezing of the spin ensemble above 400 mT. Moreover, we find the qubit to be coupled to a recently discovered, long-lived TLS ensemble[34,47], which accounts for half of the dissipation budget. Notably, we do not observe a magnetic dependence of this coupling, challenging the recently proposed spins hypothesis as its origin[34]. Finally, we show that the

critical current noise reported in ref. 46 is not magnetic field susceptible.

## Results

In Fig. 1 we present the field resilient gralmonium qubit, fabricated from a single layer of grAl (cf. Fig. 1a), with a critical field on the order of $B_c \sim 6\,T$[39]. We use a 20 nm thick grAl film with a sheet inductance of $L_\square = 0.75\,nH/\square$ (resistivity $\rho = 2000\,\mu\Omega\,cm$) to design all circuit elements (cf. "Methods"). We galvanically couple a 1 mm long stripline readout resonator to the qubit circuit, consisting of a superinductor, a geometric finger capacitance and a grAl nanojunction. Implemented by a ~(20 nm)³ grAl volume, the nanojunction offers a sinusoidal current-phase relation similar to conventional Al/AlO$_x$/Al JJ[46], while exposing a minute cross-section to Fraunhofer interference. To reduce the sensitivity to magnetic flux fluctuations perpendicular to the thin film, we implement a gradiometric design[48] with two flux loops (ocher & violet in Fig. 1a) containing fluxes $\Phi_1$, $\Phi_2$, respectively. The equivalent circuit diagram in Fig. 1b can be mapped to the standard fluxonium Hamiltonian

$$H = 4E_C \hat{n}^2 + \frac{1}{2}E_L\left(\hat{\varphi} - 2\pi\frac{\Phi_{ext}}{\Phi_0}\right)^2 - E_J\cos\hat{\varphi}, \quad (1)$$

where $E_L = (\Phi_0/2\pi)^2/L_q$, $E_C = e^2/2C$, $E_J = I_c\Phi_0/2\pi$ and $\Phi_0 = h/2e$. Here, $\hat{n}$ represents the number of Cooper pairs and $\hat{\varphi}$ is the phase difference

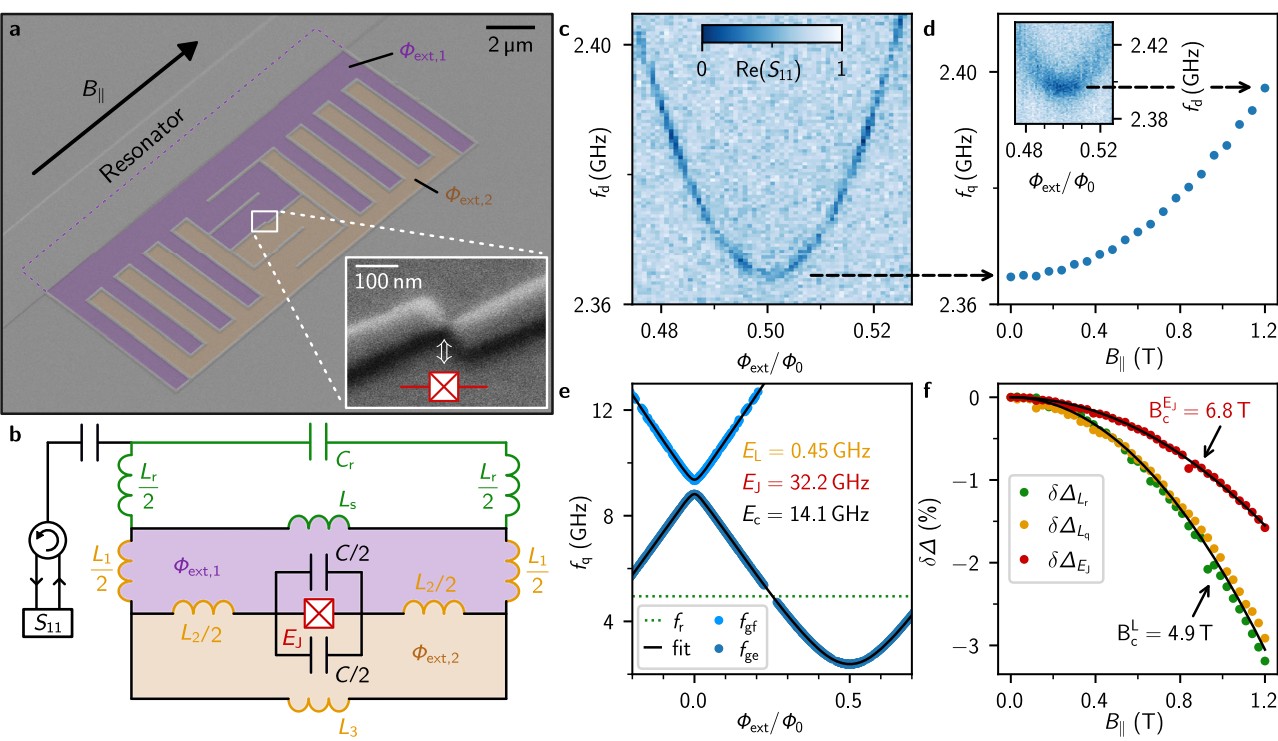

**Fig. 1 | Gradiometric gralmonium qubit resilient to Tesla magnetic field. a** False-colored scanning electron microscope (SEM) image of the qubit circuit, galvanically coupled to the readout resonator. The device consists of a 20 nm thick single layer of grAl. The colored regions (ocher & violet) illustrate the 10% mismatched areas of the two flux loops in the gradiometric design[48], which result in an effective flux bias $\Phi_{ext}$ in perpendicular magnetic field $B_\perp$ (cf. Eq. (2)). Inset: zoom-in on the ~20 nm wide grAl nanojunction of the qubit (cf. ref. 46). **b** Circuit schematic for the gradiometric qubit depicted in **a**: the nanojunction (red) is shunted by an inter-digitated capacitor and two flux loops (ocher & violet) with inductances $L_1 + L_s$ and $L_3$, respectively. The inductance shared between the loops is $L_2$. The qubit is inductively coupled via $L_s$ to the readout resonator (inductance $L_r$, capacitance $C_r$) for which we measure the single-port reflection coefficient $S_{11}$. **c** Two-tone (TT) spectroscopy at the half flux sweet spot $\Phi_{ext} = \Phi_0/2$ in $B_\parallel = 0\,T$. **d** Increase of the

sweet spot qubit frequency in magnetic field up to 1.2 T. Inset: TT-spectroscopy in $B_\parallel = 1.2\,T$. **e** Qubit spectrum: ground to excited ($f_{ge}$ in dark blue markers) and ground to second-excited ($f_{gf}$ in light blue markers) state transitions extracted from TT-spectroscopy. From a fit (black line) to the fluxonium Hamiltonian (Eq. (1)), we obtain $E_J/h = 32.2\,GHz$ (i.e., critical current $I_c = 64.9\,nA$), $E_C/h = 14.1\,GHz$ ($C = 1.37\,fF$) and $E_L/h = 0.454\,GHz$ ($L_q = 360\,nH$). **f** Suppression of the grAl superconducting gap $\Delta$ in magnetic field. The red and orange markers, corresponding to the qubit nanojunction and inductor superconducting gaps ($\Delta_{E_J}$, $\Delta_{L_q}$), are obtained from fitted $E_J$ and $E_L$ values (cf. **e**) at each magnetic field. The capacitance $C$ is fixed to the fit value obtained in $B_\parallel = 0\,T$. The green markers are obtained from the shift of the readout resonator frequency $f_r(B_\parallel)$. The black lines show fits to the field dependence of the superconducting gap, indicating a 40% higher critical field for the nanojunction.

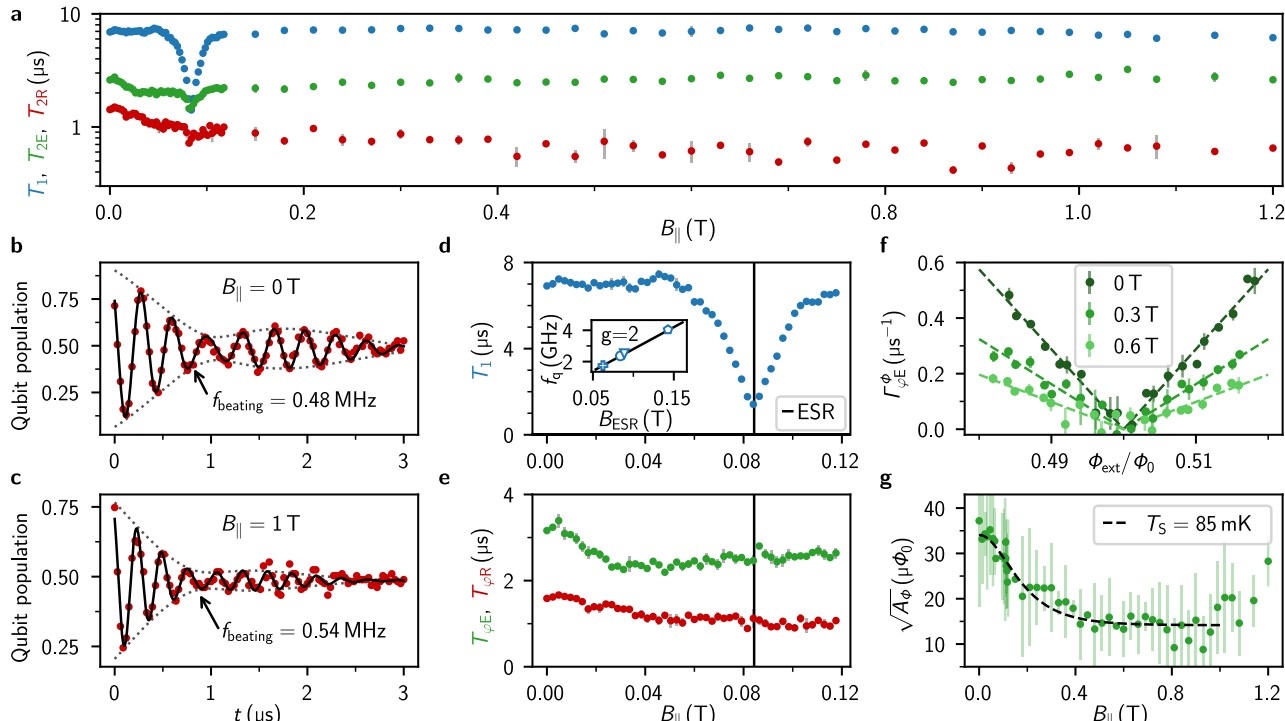

**Fig. 2 | Qubit coherence in magnetic field: signatures of environmental spin polarization.** **a** Energy relaxation time $T_1$, Ramsey and echo coherence time, $T_{2R}$ and $T_{2E}$ respectively, of the gradiometric gralmonium in magnetic field up to 1 T. **b**, **c** Ramsey fringes measured in $B_\parallel = 0$ T and $B_\parallel = 1$ T, respectively. A two-frequency fit (black line) indicates a similar beating pattern (dotted envelope) for both magnetic fields. **d** Energy relaxation $T_1$ up to 120 mT: similarly to observations on resonators[39-41], the drop in $T_1$ suggests coupling to the electron spin resonance (ESR) of paramagnetic impurities of unknown origin. Inset: The fields $B_{ESR} = hf_q/g\mu_B$ at which the ESR matches different qubit frequencies in different cooldowns,

correspond to the expectation for a spin $s = 1/2$ ensemble with gyromagnetic factor $g = 2$ (black line). Note that we use the same device for which the qubit frequency changes between cooldowns (cf. ref. 46). **e** Dephasing times $T_{\varphi R}$, $T_{\varphi E}$ remain unaffected by the ESR. **f** Flux noise echo dephasing rate $\Gamma_{\varphi E}^\Phi$ in the vicinity of the sweet spot for three in-plane magnetic fields. Dashed lines show fits to Eq. (3). **g** Flux noise amplitude $\sqrt{A_\Phi}$ in magnetic field with fit to Eq. (4), corresponding to a spin freezing with a spin temperature of $T_S = 85$ mK. In all panels, the error bars represent the standard deviation obtained from successive measurements.

across the junction. Due to the low intrinsic capacitance of the nanojunction, the qubit charging energy $E_C$ is dominated by the interdigitated capacitor $C$[46]. For the gradiometric circuit, the effective qubit inductance is given by $L_q = \frac{L_{1,s}L_2 + L_2L_3 + L_3L_{1,s}}{L_{1,s} + L_3}$ with $L_{1,s} = L_1 + L_s$, and the effective external flux is

$$\Phi_{ext} = \Phi_\Delta - \alpha\Phi_\Sigma. \qquad (2)$$

Here, $\Phi_{\Sigma/\Delta} = \frac{\Phi_{ext,1}}{2} \pm \frac{\Phi_{ext,2}}{2}$ denote the mean and difference of fluxes in the two loops, respectively, and $\alpha = \frac{L_{1,s} - L_3}{L_{1,s} + L_3}$ is the inductance asymmetry. In our gradiometric design, the magnetic flux susceptibility is reduced by a factor of $\Phi_{ext,1}/\Phi_\Delta = 4.6$ with $\alpha \approx 0$ (cf. Supplementary II).

From two-tone (TT) spectroscopy at half flux bias $\Phi = \Phi_0/2$ shown in Fig. 1c, we determine a qubit frequency of $f_q(\Phi_0/2) = 2.365$ GHz in zero field, $B_\parallel = 0$. Remarkably, as shown in Fig. 1d, tracking the sweet spot qubit frequency in magnetic field reveals only a 1% increase (32 MHz) up to 1.2 T, illustrating the compatibility of the gradiometric gralmonium qubit with high magnetic fields. The spectroscopy data in 1.2 T is blurred compared to zero field due to low-frequency flux noise, likely from vibrations of the sample holder inside the vector magnet (cf. Supplementary I). Figure 1e shows the gralmonium spectrum up to 13 GHz, extracted from TT spectroscopy. A joint fit of the qubit transitions $|g\rangle \rightarrow |e\rangle$ and $|g\rangle \rightarrow |f\rangle$ to a numerical diagonalization of Eq. (1) yields typical fluxonium parameters: $E_J/h = 32.2$ GHz, $E_C/h = 14.1$ GHz and $E_L/h = 0.454$ GHz.

To assess the effect of the magnetic field on the fluxonium parameters, we measure the qubit ground to excited transition frequency

$f_{ge}$ near the half- and zeroflux sweet spots at each $B_\parallel$, using TT spectroscopy (similar to Fig. 1c). We fit $f_{ge}$ to Eq.(1) using the field independent capacitance $C = 1.37$ fF obtained in zero field. From the fitted parameters, using $E_J \propto \Delta(B_\parallel)$ and $L_{kin} \propto 1/\Delta(B_\parallel)$, we extract the magnetic field suppression of the superconducting gap in the superinductor ($\delta\Delta_{L_q}$) and nanojunction ($\delta\Delta_{E_J}$), as shown in Fig. 1f. The suppression of the resonator superconducting gap ($\delta\Delta_{L_r}$) is independently obtained by fitting the resonance frequency shift $f_r \propto 1/\sqrt{L_{kin}}$. Interestingly, the nanojunction has an even higher field resilience than the grAl resonator and superinductor. We fit the relative gap suppression to $\sqrt{1 - (B_\parallel/B_c)^2}$[49] and obtain a critical field $B_c^{E_J} = 6.8$ T for the nanojunction and $B_c^L = 4.9$ T for the resonator and qubit inductance. The fact that $B_c^{E_J} > B_c^L$ indicates that possible Fraunhofer interference in the nanojunction plays a minor role. The higher critical field of the nanojunction is not understood and could be due to its reduced dimensions, similar to ref. 50.

We quantify the quantum coherence of the gralmonium in field by performing time-domain measurements at the half flux sweet spot, as summarized in Fig. 2. Remarkably, the energy relaxation time $T_1$ and Hahn echo coherence time $T_{2E}$ remain robust in fields up to 1.2 T, the upper limit of our vector magnet (cf. Fig. 2a). The Ramsey coherence time $T_{2R}$ decreases from a maximum of $T_{2R} = 1.5$ μs to $T_{2R} = 0.7$ μs in fields above 1 T. We attribute this to an increase of low frequency flux noise, which stems from global flux variations introduced by vibrations and activated vortices in the vector magnet[51], or from local flux noise, possibly from spins clusters[38].

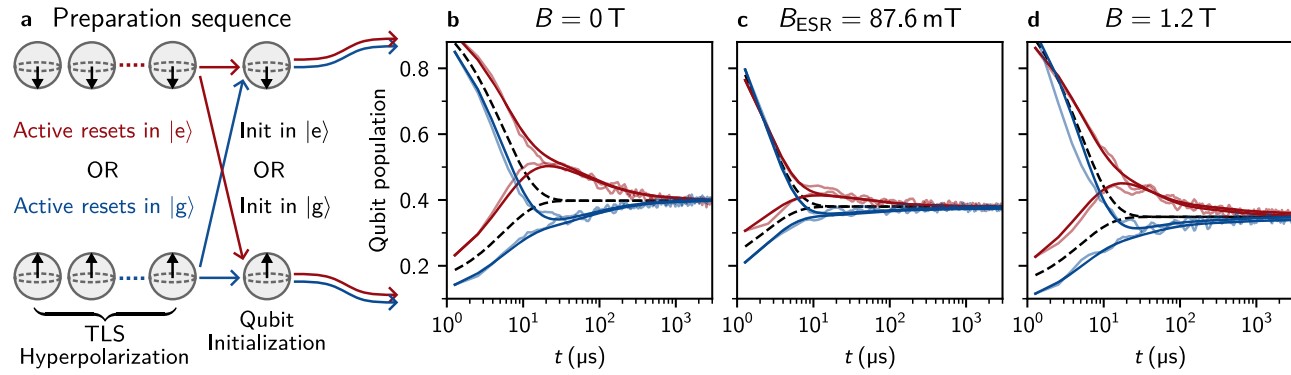

**Fig. 3 | Magnetic susceptibility of long-lived two-level-systems (TLSs) in high field. a** Sketch of the qubit preparation sequence used in (**b**–**d**). The repeated ($N = 10^4$) active reset of the qubit state in $|g\rangle$ or $|e\rangle$ (blue and red traces in all panels, respectively) results in the hyperpolarization of environmental, long-lived TLS[34]. The last step of the preparation sequence consists in a qubit initialization in $|g\rangle$ or $|e\rangle$. We use a 540 ns rectangular readout pulse and a 32 ns Gaussian $\pi$-pulse. **b** Qubit population relaxation after the preparation sequence for different magnetic fields $B_{\parallel}$. We fit the data (semi-transparent) to the theoretical model[34,55] (opaque). For reference, the black dashed lines show an exponential decay with the qubit energy relaxation rate $\Gamma_1$. In zero field, we reproduce the signatures of TLS hyperpolarization recently observed in other superconducting qubits[34,35], i.e., undershoot (blue) and overshoot (red) compared to the single exponential decay. **c** At the ESR resonance field $B_{ESR}$, the hyperpolarization signatures are suppressed due to energy relaxation of the qubit into the paramagnetic ensemble. **d** The signatures of TLS hyperpolarization on qubit relaxation in magnetic fields exceeding 1 T are comparable to zero field, indicating a very low susceptibility of the long-lived TLSs to magnetic field.

The grAl nanojunction exhibits critical current fluctuations, as evidenced by a 0.5 MHz toggling of the qubit frequency and a corresponding beating pattern in Ramsey fringes at zero field (cf. Fig. 2b). As demonstrated in ref. 46, these fluctuations are inconsistent with transverse coupling to a fixed frequency TLS, but originate from fluctuations of the nanojunction energy, potentially arising from structural defects, charge noise, or paramagnetic impurities. This issue is also relevant for standard Al/AlO$_x$/Al tunnel JJs[30,52]. We leverage the gradiometric gralmonium's field resilience to test the magnetic susceptibility of critical current noise, showing in Fig. 2c that a 1 T in-plane magnetic field does not suppress the discrete fluctuations of the Josephson energy. This observation excludes magnetically susceptible sources, such as a local spin environment, as the origin of these fluctuations. Further experiments, such as electric field bias or mechanical strain on the substrate[32,53] or spin-locking TLS spectroscopy[54], are required to identify their cause.

We observe a dip in the energy relaxation time at the magnetic field where the electron spin resonance (ESR) matches the qubit frequency hf = $g\mu_B B$ (cf. Fig. 2d). This ESR resonance does not impact the dephasing times $T_{\varphi R}$, $T_{\varphi E}$ (cf. Fig. 2e), as expected in the limit of a coupling strength much smaller than the qubit linewidth[55]. By exploiting $\Delta E_J$ - GHz changes in the nanojunction energy after thermal cycling, we change the qubit frequency of the same device, allowing measurements of the ESR-resonant field $B_{ESR}$ across multiple qubit frequencies at the half flux sweet spot (inset of Fig. 2d). The linear trend of the extracted magnetic fields $B_{ESR}$ aligns with the prediction for a $g = 2$ spin $s = 1/2$ ensemble, identifying a paramagnetic spin ensemble coupled to our qubit.

Figure 2f illustrates the flux dependence of the Hahn echo flux noise dephasing rate, $\Gamma_{\varphi E}^{\Phi}$, near the sweet spot for three in-plane magnetic fields ($B_{\parallel} = 0$, $B_{\parallel} = 0.3$ T, $B_{\parallel} = 0.6$ T). Away from the sweet spot, we observe a Gaussian contribution in the Hahn echo decay curve, consistent with commonly observed 1/$f$ flux noise[27–31,38,56]. We extract $\Gamma_{\varphi E}^{\Phi}$ from the flux-dependent Gaussian envelope $e^{-(\Gamma_{\varphi E}^{\Phi} t)^2}$, on top of a purely exponential decay $e^{-(\Gamma_1/2 + \Gamma_{\varphi E}^{const})t}$, extracted at $\Phi = \Phi_0/2$. The flux-independent term $\Gamma_{\varphi E}^{const}$ may originate from critical current noise or photon shot noise; in the latter case, the residual photon number is $\bar{n} = 0.27$, corresponding to an effective temperature of 150 mK, in agreement with the qubit temperature (cf. Supplementary I). Interestingly, the flux dependence $\Gamma_{\varphi E}^{\Phi}(\Phi_{ext})$ weakens as the magnetic field $B_{\parallel}$ increases (cf. Fig. 2f), reminiscent of earlier observations

in flux qubits at lower field[38]. We fit the flux noise amplitude $\sqrt{A_\Phi}$ for a $S_\Phi(\omega) = A_\Phi/\omega$ power spectral density using[29,56].

$$\Gamma_{\varphi E}^{\Phi} = \sqrt{A_\Phi \ln 2} \left| \frac{\partial \omega}{\partial \Phi_{ext}} \right|. \tag{3}$$

With increasing $B_{\parallel}$, we observe a decrease of $\sqrt{A_\Phi}$ by a factor of ~2, reported in Fig. 2g, which holds across different qubit frequencies in several cooldowns (cf. Supplementary III). However, for $B_{\parallel} \gtrsim 1$ T, $\sqrt{A_\Phi}$ increases, suggesting the onset of a competing mechanism, likely due to vortex dynamics in the magnet wires.

We model the flux noise as the sum of a large number of magnetic two-level fluctuators, consistent with the commonly accepted spin-based origin of flux noise[57–59]. Each of them constitutes a source of asymmetric random telegraphic noise, with a Lorentzian power spectrum $S(\omega) \propto (\Gamma_1/\Gamma_\uparrow + \Gamma_1/\Gamma_\downarrow)^{-1} \cdot \Gamma_1/(\Gamma_1^2 + \omega^2)$, where $\Gamma_1 = \Gamma_\uparrow + \Gamma_\downarrow$ are the excitation and relaxation rates of the fluctuator, respectively[60]. In the limit of identical fluctuators, $S(\omega)$ remains Lorentzian, while for fluctuators with $1/\Gamma_1$ uniformly distributed, $S(\omega) \propto 1/\omega$[56]. However, for any distribution, the amplitude of the power spectrum is $A_\Phi \propto (\Gamma_1/\Gamma_\uparrow + \Gamma_1/\Gamma_\downarrow)^{-1}$, which becomes (cf. Supplementary III).

$$A_\Phi \propto 1/\cosh^2\left(\frac{\mu_B B}{k_B T_S}\right). \tag{4}$$

Here, $2\mu_B B$ is the energy of $g = 2$, $s = 1/2$ paramagnetic impurities and $\mu_B$, $k_B$ and $T_S$ are the Bohr magneton, the Boltzmann constant and the spin bath temperature, respectively. A fit with $T_S = 85$ mK aligns with the measured flux noise amplitude (black line in Fig. 2g) up to 1 T. This suggests the freezing of $g = 2$ paramagnetic impurities responsible for the reduction of flux noise, presumably the same spin environment that causes the $T_1$ dip (cf. Fig. 2d).

In Fig. 3, we leverage the field resilience of the gralmonium to probe the magnetic susceptibility of a recently discovered TLS bath coupled to superconducting qubits[34,47,55]. These TLSs have been shown to induce non-Markovian qubit dynamics, and their long lifetime, exceeding 1/$\Gamma_{TLS} \gtrsim 50$ ms, makes a spin-based origin plausible. Following ref. 34, by repeatedly preparing the qubit in either $|g\rangle$ or $|e\rangle$ using fast feedback over $N = 10^4$ iterations, the TLS ensemble hyperpolarizes via its cross-relaxation to the qubit. After this polarization sequence, the qubit is initialized in either $|g\rangle$ or $|e\rangle$, and its population is

monitored using stroboscopic quantum jump measurements. Figure 3b shows the distinct signatures of a hyperpolarized long-lived TLS ensemble coupled to the gralmonium: regardless of the qubit's initial state, it relaxes to the TLS ensemble population on a $T_1$ timescale, while the TLS ensemble itself decays to thermal equilibrium on milliseconds timescale. By modeling the qubit coupled to a ladder of $10^2$ TLSs[34], we extract a gralmonium relaxation $\Gamma_1 = 1/5.4\,\mu s$, of which TLS cross-relaxation accounts for $\sum_k \Gamma_{qt}^k = 1/22\,\mu s$.

In magnetic field, the signatures of TLS hyperpolarization remain visible, as illustrated in Fig. 3c, d. The fact that the hyperpolarization in $B_\parallel = 1.2\,T$ is comparable to zero field indicates that the TLS bath is not magnetically susceptible, ruling out origins, such as electronic spins. Remaining non-magnetically-susceptible microscopic origins include subgap states, possibly trapped quasiparticles[61]. As shown in Fig. 3c, at $B \approx B_{ESR}$, where $T_1$ is suppressed by a factor of 7 (cf. Fig. 2d), the TLS hyperpolarization is less pronounced. Therefore, we are still able to hyperpolarize the long-lived TLSs, but not the paramagnetic spins. This indicates that the spin ensemble is large enough or sufficiently coupled to the environment that it embodies a Markovian bath. In contrast, the long-lived TLS environment appears to be uncoupled to the spin ensemble, as evidenced by the fit in Fig. 3c with a practically unchanged cross-relaxation rate of $\sum_k \Gamma_{qt}^k = 1/33\,\mu s$.

In conclusion, we have introduced a field-resilient superconducting qubit—the gradiometric gralmonium—that operates robustly in Tesla magnetic fields. By incorporating a grAl nanojunction, the gralmonium maintains spectral stability and coherence in high magnetic fields, circumventing the Fraunhofer interference typically observed in JJ-based superconducting circuits. We reveal distinct properties of spin environments coupled to the gralmonium by addressing their magnetic field susceptibility. Using ESR, we characterize a paramagnetic spin-1/2 ensemble that couples transversely to the qubit, demonstrating the gralmonium's potential as a probe for spin dynamics. We confirm the long-standing hypothesis of the freeze-out of fast flux noise in high fields, consistent with a spin $s = 1/2$, $g = 2$ paramagnetic origin. The operation of the gralmonium in magnetic field also allowed us to disprove the electron-spin hypothesis for the long-lived two-level system (TLS) environment responsible for non-Markovian qubit dynamics.

Future work should address flux noise suppression and spectral noise analysis, and should validate the correlation between flux noise and the spin-1/2 ensemble in order to gain insights into its microscopic origin. Most importantly, the gralmonium's resilience to magnetic fields offers a promising path forward in hybrid quantum architectures, facilitating seamless integration with magnetic-field-sensitive systems, such as spins[62], magnons, or topological materials.

## Methods

The sample analyzed in this manuscript is fabricated on a double-side polished c-plane sapphire substrate using lift-off electron-beam lithography. A single resist layer of PMMA A4, coated with an 8 nm aluminum anti-static layer, is patterned with a 100 keV electron-beam writer. After patterning, the anti-static layer is removed using MF319 developer, which contains tetramethylammonium hydroxide, followed by development of the PMMA resist in a 6 °C isopropyl alcohol (IPA)/$H_2O$ solution (1:3 volume ratio). Prior to metal deposition, the substrate undergoes a 15 s $Ar/O_2$ plasma cleaning process using a Kaufman ion source. A 20 nm grAl layer is then deposited in a single evaporation step at room temperature, using an oxygen atmosphere at a chamber pressure of $\sim 1 \times 10^{-4}$ mbar and a deposition rate of approximately $1\,nm\,s^{-1}$. A titanium gettering step is performed beforehand to enhance the vacuum quality to $\sim 1 \times 10^{-8}$ mbar before evaporation. In the lift-off process, the sample is sequentially submerged in an acetone bath, a 30 min N-ethyl-2-pyrrolidone bath with ultrasonic cleaning, and finally an ethanol bath. The final sample has a sheet resistance of $1\,k\Omega/\square$.

## Data availability

All data supporting the findings of this study are available in ref. 63.

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

## Acknowledgements

We are grateful to A. Shnirman and P. Bertet for fruitful discussions and we acknowledge technical support from S. Diewald, M.K. Gamer, and L. Radtke. Funding was provided by the German Research Foundation (DFG) via the Gottfried Wilhelm Leibniz-Award (ZVN-2020_WE 4458-5) and by the European Research Council via project number 101118911 (DarkQuantum). N.Z. acknowledges funding from the Deutsche For-schungsgemeinschaft (DFG-German Research Foundation) under project number 450396347 (GeHoldeQED). M.F. acknowledges funding from the European Union under the Horizon Europe Program, grant agreement number 101080152 (TruePA). N.G and M.S. acknowledge support from the German Ministry of Education and Research (BMBF) within the project GEQCOS (FKZ: 13N15683). S.Ge. acknowledges support from the German Ministry of Education and Research (BMBF) within the project QSolid (FKZ: 13N16151). Facilities use was supported by the Karlsruhe Nano Micro Facility (KNMFi) and KIT Nanostructure Service Laboratory (NSL). We acknowledge qKit for providing a convenient measurement software framework.

## Author contributions

S.Gü., J.B., D.R., and I.M.P. conceived and designed the experiment. S.Gü., J.B., J.K.H., and A.B. fabricated the device. S.Gü., J.B., D.R., N.G., N.Z., M.F., S.Ge., and M.S. participated in the measurements. S.Gü., J.B., N.G., and M.S. analyzed the data. S.Gü. and J.B. led the paper writing, and all authors contributed to the text. W.W. and I.M.P. supervised the project.

## Funding

## Competing interests

The authors declare no competing interests.
