## [Transparent Peer Review File · Nature Communications]

Spin Environment of a Superconducting Qubit in High Magnetic Fields

Corresponding Author: Mr Simon Günzler

Version 0:

Reviewer comments:

Reviewer #1

(Remarks to the Author)

The manuscript by Günzler et al. explores the performance of a specific superconducting qubit circuit, the “grAlmonium” which is a fluxonium made with granular aluminum superinductor and nanoconstriction Josephson junction, as a function of applied magnetic field. The authors identify coupling to $g=2$ spins via an electron-spin-resonance-like measurement that uses the relaxation time of the fluxonium qubit. They see a suppression of the flux noise amplitude with field, seeming consistent with the expectation of spin freezing. Furthermore, they explore the TLS environment of the qubit through a hyperpolarization experiment, discussed in their Ref. 34, finding the ability to hyperpolarize the TLS defects even at high fields. From this they conclude that the TLS bath is not susceptible to magnetic fields and is likely distinct from the paramagnetic spin ensemble – a conclusion that is consistent with the current understanding in the community. As such, it is not exactly surprising, but it’s an interesting datapoint toward understanding the microscopic origins of TLS. Aside from a few comments that should be addressed, I feel the work is technically sound and warrants publication.

- Line 47 – my understanding is that those junctions are still susceptible to Fraunhofer, just in a different field orientation? For nanoconstriction junctions, is it still correct to think that one would only be sensitive to Fraunhofer from fields perpendicular to the current flow? How would that look given the authors’ model of granularity (a network of JJs) giving rise to kinetic inductance in this material?
- Fig. 1f: Related to the above, do the authors know for sure that the EJ suppression is due primarily to superconducting gap? Is it inconsistent with Fraunhofer reduction given their junction area? Granted, the suppression of resonator frequency likely supports this too, but curious if this was checked.
- Fig 2d: The observed ESR dip in T1 is promising, indicating that they are indeed coupled to a native paramagnetic spin ensemble. They mention that they don’t see this dip in T_{ϕ_R} or T_{ϕ_E} , and they point out this is consistent in the limit of coupling strength much smaller than the qubit linewidth. But it seems natural not to see it because T_{ϕ_R} and T_{ϕ_E} are sensitive to frequencies significantly lower than the Zeeman splitting. That said, it’s interesting that the low-frequency noise is not impacted by high fields, at least within the resolution they capture.
- Line 67 – also some work from Chen Wang’s group to cite
- Eq1 – for operators, add hats or make the text bold
- Line 153 – this frequency toggling could be explained by dispersive coupling to a TLS, correct? And not only the critical current noise that the authors suggest? I take issue with their choice to call these “critical current” fluctuations throughout the paragraph, as it could arise from charge coupling.
- Line 175: Why does the nanojunction energy change so dramatically between thermal cycles? Can they comment on whether and/or why this is normal in a grAl qubit?
- Fig. 7: Any comment on why the fitted spin bath temperatures (T_s) are so different between thermal cycles / qubit frequencies?
- Line 192 – do the authors fit their exponential fit parameter to be equal to Γ_1 as measured from an energy relaxation measurement, or leave the parameter free? There can be other exponential contributions to dephasing beyond just T1 arising from coupling to TLS.
- Line 285 – the word “infirm” should be changed for clarity
- Fig. 2a/e: 2a and 2e plot T2 and T_{ϕ} , respectively. Perhaps it would be more instructive to plot T_{ϕ} in both, where it would also be apparent that the ESR dips only appear in T1 and the T1 contribution to T2. But this is not critical
- Some additional description of the hyperpolarization experiment in this text as opposed to relying on references would be beneficial for the reader.

Reviewer #2

(Remarks to the Author)

Summary

This work presents a superconducting fluxonium qubit based on granular aluminum (grAl), which operates stably in magnetic fields exceeding 1 Tesla. By combining a superinductance and an ScS-type junction—both fabricated from disordered materials—the authors develop a qubit platform that avoids Fraunhofer interference and maintains minimal frequency shifts under strong in-plane magnetic fields. The magnetic-field resilience enables a systematic study of the qubit's spin environment. The authors demonstrate that most of the flux noise originates from a paramagnetic spin- $1/2$ bath that "freezes out" at high field, as evidenced by a significant reduction in the $1/f$ noise amplitude via Hahn echo dephasing. In contrast, a previously observed long-lived TLS bath—responsible for non-Markovian dynamics and hyperpolarization effects—remains insensitive to magnetic fields, suggesting a non-magnetic origin, likely related to amorphous dielectric charge defects. These findings offer new experimental support for two long-standing hypotheses: that fluctuating spin baths cause flux noise, and that charge-based TLSs are responsible for dielectric losses in superconducting qubits.

Significance and Validity

Understanding the microscopic origins of decoherence remains a major challenge in the field of superconducting quantum devices. While environmental degrees of freedom degrade qubit coherence, they can also be probed using qubit-based noise spectroscopy. Among the two most generic noise sources—flux and charge noise—flux noise is particularly detrimental to flux-type qubits like persistent current qubit and fluxonium and is believed to arise from fluctuating spins near the superconducting circuit. Studying the magnetic susceptibility of this spin bath is thus a natural strategy for identifying its microscopic origin.

However, the incompatibility between superconducting circuits and magnetic fields has made in situ probing of spin baths under strong fields extremely challenging. The authors overcome this limitation using a previously developed magnetic-field-resilient fluxonium device ("gralmonium"). Notably, the qubit parameters remain largely unchanged, and coherence times—while moderate by current standards—are preserved up to 1 Tesla.

The authors employ a comprehensive suite of experimental techniques, including Hahn echo, Ramsey, T_1 measurements, and environmental hyperpolarization via active qubit resets. They verify the $g = 2$ nature of the spin bath through ESR signatures across cooldowns with varying qubit frequencies. They also successfully disentangle spin-induced noise (flux noise) from TLS-induced effects (dielectric loss), using magnetic field dependence as the distinguishing factor.

One important inconsistency, however, arises in the reported effective temperatures. The qubit temperature inferred from thermal population is ~ 165 mK—substantially higher than both the cryostat temperature (~ 30 mK) and the ~ 37 mK value reported in Ref. 46 using a similar device and setup. Furthermore, the fitted spin bath temperature ($T_s = 85$ mK) differs significantly from the qubit temperature. Across cooldowns, T_s ranges from 61 to 298 mK without clear explanation. This discrepancy raises questions about the accuracy of the methodology and suggests the potential for alternative interpretations beyond spin bath freezing due to magnetic polarization.

Technical Questions and Comments for Improvement

1. Abstract – "Suppression of fast flux noise": The term "fast flux noise" is vague. Since noise at different frequencies may have different origins, the specific frequency range under investigation should be clearly stated.
2. Line 21–27 – Some references (12–19) involve spin qubits coupled to superconducting resonators rather than qubits. These are mainly relevant to spin qubit readout and less connected to the topic of flux noise in superconducting qubits. Consider replacing them with more relevant works.
3. Literature Context – The manuscript would benefit from a broader contextualization of prior flux noise studies in superconducting qubits. Notable references include:
 - Quintana et al., Phys. Rev. Lett. 118, 057702 (2017): Noise spectrum over a broad frequency range.
 - Lanting et al., Phys. Rev. B 79, 060509(R) (2009): Geometric dependence of low-frequency flux noise.
 - Gao et al., Nat. Commun. 16, 3620 (2025): Material dependence of flux noise. These references reinforce the significance of identifying spin noise origins and would enhance the motivation for the high-field measurements.
4. Figure 1f – Clarify how qubit parameters at different magnetic fields were extracted. Were qubit spectra measured as a function of flux bias at each temperature? How are L_q and L_r independently determined?
5. Lines 153–158 – The observed Ramsey fringe beating is attributed to critical current fluctuations. Abdurakhimov et al., PRX Quantum 3, 040332 (2022), report a type of TLS affecting critical currents—distinct from standard charge TLSs—that may explain these observations. Please discuss this possibility.
6. Lines 234–235 – The statement that long TLS lifetimes imply a spin environment is questionable. Charge-based TLSs can also exhibit long lifetimes depending on tunneling amplitude.
7. Line 250 – The cross-relaxation rates used to model TLS hyperpolarization should be clearly described. Given that this technique is relatively new, please include the theoretical model and fitting procedure to make the manuscript self-contained.
8. Lines 631–634 – The high thermal population is attributed to missing thermal shielding. However, this contradicts the claim that the setup is identical to Ref. 46, which reported significantly lower effective temperatures. Please clarify this discrepancy.

Reviewer #3

(Remarks to the Author)

Version 1:

Reviewer comments:

Reviewer #2

(Remarks to the Author)

Overall, the authors have made a commendable effort to address the referees' concerns, and the clarity of the manuscript has significantly improved.

One important limitation that remains unresolved is the large variation in the extracted spin-bath temperatures across different cooldowns. The authors argue that these discrepancies should not be overinterpreted, as they result from fits to flux-noise amplitudes rather than direct spectral measurements. I find this reasoning unconvincing. By definition, T_s is the characteristic temperature at which the polarization energy of the field exceeds thermal spin fluctuations, and it should therefore reflect the effective spin-bath temperature. The reliability of this quantity is not determined by whether it is obtained from direct spectral measurements—which in any case are unlikely to yield it. I do appreciate the experimental difficulty of achieving low effective temperatures in a setup with a vector magnet. Nevertheless, I strongly recommend that the manuscript explicitly acknowledge the limitation posed by the large variation in extracted spin-bath temperatures, so that readers can make their own informed judgment when interpreting the data.

Despite this limitation, the study provides valuable new decoherence measurements under applied magnetic fields, offering experimental evidence that supports two long-standing decoherence mechanisms in superconducting qubits. On this basis, I find the work of broad interest and recommend publication in Nature Communications.

Reviewer #3

(Remarks to the Author)

We would like to start by thanking both reviewers for their words of appreciation and their interest in the manuscript. We also welcome their constructive criticism, which we feel significantly improved the clarity of the manuscript.

In the following we offer a point-by-point reply and we describe the corresponding changes to the manuscript. The reviewers' comments are listed in blue and our response in black. The modifications to the manuscript are highlighted in the PDF.

Reviewer 1:

The manuscript by Günzler et al. explores the performance of a specific superconducting qubit circuit, the "grAlmonium" which is a fluxonium made with granular aluminum superinductor and nanoconstriction Josephson junction, as a function of applied magnetic field. The authors identify coupling to $g=2$ spins via an electron-spin-resonance-like measurement that uses the relaxation time of the fluxonium qubit. They see a suppression of the flux noise amplitude with field, seeming consistent with the expectation of spin freezing. Furthermore, they explore the TLS environment of the qubit through a hyperpolarization experiment, discussed in their Ref. 34, finding the ability to hyperpolarize the TLS defects even at high fields. From this they conclude that the TLS bath is not susceptible to magnetic fields and is likely distinct from the paramagnetic spin ensemble – a conclusion that is consistent with the current understanding in the community. As such, it is not exactly surprising, but it's an interesting datapoint toward understanding the microscopic origins of TLS. Aside from a few comments that should be addressed, I feel the work is technically sound and warrants publication.

We thank the reviewer for their positive evaluation and in the following we address their specific questions.

- Line 47 – my understanding is that those junctions are still susceptible to Fraunhofer, just in a different field orientation? For nanoconstriction junctions, is it still correct to think that one would only be sensitive to Fraunhofer from fields perpendicular to the current flow? How would that look given the authors' model of granularity (a network of JJs) giving rise to kinetic inductance in this material?

The reviewer is correct that nanoconstriction part of the nano-JJ remains in principle susceptible to Fraunhofer interference from magnetic fields perpendicular to the current flow. For our $\sim 20 \times 20$ nm² cross-section, the maximum calculated suppression assuming the field is perpendicular to the current through the nano-JJ, is $1 - I_c(B=1T)/I_c(B=0) = 6\%$. Moreover, the granular microstructure of grAl introduces significant kinetic inductance in the path of the persistent currents through the nano-JJ, further reducing the possible Fraunhofer suppression. As a result, we don't expect the nano-JJ critical current suppression to be more susceptible to field compared to the resonator and the superinductor. What is strikingly surprising in our experiments is that the nano-JJ is actually more resilient to magnetic field compared to the resonator and the superinductor, which allows us to dismiss a significant contribution from the Fraunhofer suppression. With this being said, the exact Fraunhofer contribution is likely not zero, but in order to extract it from the measured data, we would first need to understand why the critical field of the nano-JJ is larger than that of the rest of the circuit, even though they are made from the same film.

Following the reviewer's remark, we rephrased the ending of the first paragraph on the second column of page 2, which now reads:

"The fact that $B_c^{EJ} > B_c^L$ indicates that possible Fraunhofer interference in the nanojunction plays a minor role. The higher critical field of the nanojunction is not understood and could be due to its reduced dimensions, similar to Ref. [50]."

- Fig. 1f: Related to the above, do the authors know for sure that the EJ suppression is due primarily to superconducting gap? Is it inconsistent with Fraunhofer reduction given their junction area? Granted, the suppression of resonator frequency likely supports this too, but curious if this was checked.

Yes, the measured nano-JJ suppression in field is inconsistent with $\sin(x)/x$ and follows $\sqrt{1-(B/B_c)^2}$ as expected from the suppression of the gap in field. See also the reply to the previous point and the associated modification in the text.

- Fig 2d: The observed ESR dip in T_1 is promising, indicating that they are indeed coupled to a native paramagnetic spin ensemble. They mention that they don't see this dip in $T_{\text{phi_R}}$ or $T_{\text{phi_E}}$, and they point out this is consistent in the limit of coupling strength much smaller than the qubit linewidth. But it seems natural not to see it because $T_{\text{phi_R}}$ and $T_{\text{phi_E}}$ are sensitive to frequencies significantly lower than the Zeeman splitting. That said, it's interesting that the low-frequency noise is not impacted by high fields, at least within the resolution they capture.

We agree with the reviewer that the absence of an ESR feature in the dephasing times is expected. Regarding the robustness to low frequency noise in high magnetic field: we believe that the gradiometric qubit design is effective at mitigating global flux fluctuations (e.g., from the vector magnet), which could otherwise become a roadblock at high field.

- Line 67 – also some work from Chen Wang's group to cite

We thank the reviewer for pointing out this recent arXiv preprint, which was posted while our manuscript was in review. We agree that this work from Chen Wang's group provides additional support for the existence of long-lived TLSs coupled to superconducting qubits even in standard Aluminum/AlOx qubits, not only in grAl. We have added the corresponding citation (cf. Ref [47]) to the introduction and to the Fig. 3 discussion.

- Eq1 – for operators, add hats or make the text bold

We appreciate the reviewer's attention to detail. We have added hats to the operators \hat{n} and $\hat{\phi}$ in Eq. (1) to clearly distinguish them as quantum operators.

- Line 153 – this frequency toggling could be explained by dispersive coupling to a TLS, correct? And not only the critical current noise that the authors suggest? I take issue with their choice to call these "critical current" fluctuations throughout the paragraph, as it could arise from charge coupling.

We thank the reviewer for bringing up this important comment. While in principle the beating pattern in the Ramsey fringes, manifesting as toggling between two distinct qubit frequencies, could also arise from dispersive coupling to a fixed-frequency two-level system (TLS), in our case we have reasons to believe it is not the case. We base our argument on the detailed analysis we presented in Ref. 46 (Rieger & Günzler et al., Nat. Mater. 22, 194 (2022)), using a similar nanojunction, which proves that critical current fluctuations are the dominant mechanism. Specifically, Fig. S10 in Ref. 46 (reprinted below) shows that the toggling persists across the entire qubit spectrum as a function of external flux. The two observed qubit branches can be fitted with the same circuit parameters, except for the Josephson energy E_J , which differs by 190 MHz. This behavior is consistent with a discrete toggling of the nanojunction critical current, and inconsistent with transverse coupling to a fixed-frequency TLS, which would not produce such a global shift across all flux values.

We have revised the text to clarify this interpretation:

“The grAl nanojunction exhibits critical current fluctuations, as evidenced by a 0.5 MHz toggling of the qubit frequency and a corresponding beating pattern in Ramsey fringes at zero field (cf. Fig. 2 b). As demonstrated in Ref. [46], these fluctuations are inconsistent with transverse coupling to a fixed frequency TLS, but originate from fluctuations of the nanojunction energy, potentially arising from structural defects, charge noise, or paramagnetic impurities.”

- Line 175: Why does the nanojunction energy change so dramatically between thermal cycles? Can they comment on whether and/or why this is normal in a grAl qubit?

Large changes in nanojunction energy across thermal cycles are a well-documented and recurring phenomenon in grAlmonium qubits, as systematically characterized in Ref. [46]. The nanojunction Josephson energy E_J was observed to fluctuate over timescales ranging from milliseconds to days, with the most pronounced variations occurring between thermal cycles (see Fig. 3e in Ref. [46], reprinted below). The physical origin of these fluctuations remains unresolved. Ref. [46] proposed several plausible mechanisms:

- (i) structural changes, such as tunneling crystalline defects, vacancies, interstitials, or adsorbed molecules;
- (ii) charge noise from fluctuations in locally trapped charges via the Aharonov-Casher effect; and
- (iii) paramagnetic defects.

In the present study, we experimentally rule out a magnetic origin for the observed E_J toggling by showing that its amplitude remains unchanged under an applied magnetic field of 1 T (see Fig. 2b,c). This result narrows the range of viable microscopic mechanisms, though the exact origin remains elusive. Further investigation—potentially involving electric field tuning, strain control, or local probe techniques—will be required to pinpoint the underlying cause.

- Fig. 7: Any comment on why the fitted spin bath temperatures (T_s) are so different between thermal cycles / qubit frequencies?

We agree that the variation in fitted spin bath temperatures across cooldowns and qubit frequencies is intriguing. However, we caution against overinterpreting these differences, as they arise from fits to extracted flux noise amplitudes rather than direct spectral measurements. In our view, these observations highlight the need for a more detailed spectral characterization of flux noise across field and temperature.

To clarify this point in the revised manuscript, we added a sentence in App. D:

"The observed variation in extracted spin bath temperatures motivates further noise characterization, including measurements at various cryostat temperatures and a comprehensive spectral analysis."

- Line 192 – do the authors fit their exponential fit parameter to be equal to Γ_1 as measured from an energy relaxation measurement, or leave the parameter free? There can be other exponential contributions to dephasing beyond just T1 arising from coupling to TLS.

We thank the reviewer for this insightful question. For each in-plane field value, we fit the 42 Hahn echo traces versus external flux using an exponential decay rate Γ_{exp} in which the exponential relaxation time is treated as a global fit parameter for the entire set. It captures both energy relaxation ($\Gamma_1/2$) and any additional exponential dephasing contributions, such as those from coupling to TLSs, critical current noise or photon shot noise. In contrast, the Gaussian dephasing component $\Gamma_{\phi E}^{\Phi}$ of the decay $\exp(-\Gamma_{exp} t) \exp(-(\Gamma_{\phi}^{\Phi} t)^2)$ is flux-dependent and fitted independently at each flux bias.

To clarify this procedure, in the manuscript we added a new figure in App. D, titled “**Flux-dependence of the Hahn echo decay**.”, which details the fit procedure and illustrates the transition from a purely exponential decay at half-flux bias to an increasing Gaussian decay contribution at $\Phi_{ext} \neq 0.5$. We describe the Echo modelling in App. D:

Figure 7 illustrates the fitting procedure for Hahn echo decay measurements as a function of external flux. At the half-flux sweet spot ($\Phi_{ext}/\Phi_0 = 0.5$), the decay is purely exponential, while deviations from this point ($\Phi_{ext}/\Phi_0 \neq 0.5$) reveal an additional Gaussian dephasing component, $\Gamma_{\phi E}^{\Phi}$. The echo decay is modeled as

$$\frac{1}{2} \exp(-\Gamma_{exp} t) \exp(-(\Gamma_{\phi}^{\Phi} t)^2) + \frac{1}{2}$$

The exponential decay rate $\Gamma_{exp} = \frac{\Gamma_1}{2} + \Gamma_{\phi E}^{const}$, combines energy relaxation and exponential dephasing, both assumed to be flux-independent over the range $\Phi_{ext} = 0.48 - 0.52 \Phi_0$. Accordingly, we perform a joint fit for every set of Hahn echo measurements vs. flux, extracting the flux-dependent Gaussian dephasing rate $\Gamma_{\phi E}^{\Phi}$ atop the fixed exponential decay envelope defined by Γ_{exp} .

- Line 285 – the word “infirm” should be changed for clarity

We agree with the reviewer and changed “infirm” to “disprove”:

“The operation of the galmonium in magnetic field also allowed us to disprove the electron-spin hypothesis for the long-lived two-level system (TLS) environment responsible for non-Markovian qubit dynamics.”

- Fig. 2a/e: 2a and 2e plot T2 and Tphi, respectively. Perhaps it would be more instructive to plot Tphi in both, where it would also be apparent that the ESR dips only appear in T1 and the T1 contribution to T2. But this is not critical

We thank the reviewer for the suggestion. After careful consideration, we decided to keep the current version because of the “name recognition” associated with the Ramsey T2, which we believe makes the text more accessible to a broader audience.

- Some additional description of the hyperpolarization experiment in this text as opposed to relying on references would be beneficial for the reader.

Thank you for the remark. We agree with the reviewer that an additional description of the hyperpolarization experiment would be beneficial to improve the clarity of the manuscript. This is also in line with a comment of reviewer 2. Following these remarks, we have added Appendix E and the newly added Figure 9 in the revised manuscript.

Reviewer 2:

Summary

This work presents a superconducting fluxonium qubit based on granular aluminum (grAl), which operates stably in magnetic fields exceeding 1 Tesla. By combining a superinductance and an ScS-type junction—both fabricated from disordered materials—the authors develop a qubit platform that avoids Fraunhofer interference and maintains minimal frequency shifts under strong in-plane magnetic fields. The magnetic-field resilience enables a systematic study of the qubit's spin environment. The authors demonstrate that most of the flux noise originates from a paramagnetic spin- $\frac{1}{2}$ bath that "freezes out" at high field, as evidenced by a significant reduction in the $1/f$ noise amplitude via Hahn echo dephasing. In contrast, a previously observed long-lived TLS bath—responsible for non-Markovian dynamics and hyperpolarization effects—remains insensitive to magnetic fields, suggesting a non-magnetic origin, likely related to amorphous dielectric charge defects. These findings offer new experimental support for two long-standing hypotheses: that fluctuating spin baths cause flux noise, and that charge-based TLSs are responsible for dielectric losses in superconducting qubits.

We thank the reviewer for the accurate summary.

Significance and Validity

Understanding the microscopic origins of decoherence remains a major challenge in the field of superconducting quantum devices. While environmental degrees of freedom degrade qubit coherence, they can also be probed using qubit-based noise spectroscopy. Among the two most generic noise sources—flux and charge noise—flux noise is particularly detrimental to flux-type qubits like persistent current qubit and fluxonium and is believed to arise from fluctuating spins near the superconducting circuit. Studying the magnetic susceptibility of this spin bath is thus a natural strategy for identifying its microscopic origin.

However, the incompatibility between superconducting circuits and magnetic fields has made in situ probing of spin baths under strong fields extremely challenging. The authors overcome this limitation using a previously developed magnetic-field-resilient fluxonium device ("gralmonium"). Notably, the qubit parameters remain largely unchanged, and coherence times—while moderate by current standards—are preserved up to 1 Tesla.

The authors employ a comprehensive suite of experimental techniques, including Hahn echo, Ramsey, T_1 measurements, and environmental hyperpolarization via active qubit resets. They verify the $g = 2$ nature of the spin bath through ESR signatures across cooldowns with varying qubit frequencies. They also successfully disentangle spin-induced noise (flux noise) from TLS-induced effects (dielectric loss), using magnetic field dependence as the distinguishing factor.

One important inconsistency, however, arises in the reported effective temperatures. The qubit temperature inferred from thermal population is ~ 165 mK—substantially higher than both the cryostat temperature (~ 30 mK) and the ~ 37 mK value reported in Ref. 46 using a similar device and setup.

We thank the reviewer for pointing out this aspect, which we agree that it deserves further attention. In short, the discrepancy between the qubit temperature inferred from thermal population (~ 165 mK) and the cryostat base temperature (~ 30 mK), as well as the lower temperature (~ 37 mK) reported in Ref. 46 can be explained by the different sample-holder environments. In contrast to the setup used in Ref. 46, our sample was now positioned within a 2D vector magnet anchored to the 4 K stage of the cryostat, without additional thermal shielding. This open configuration is necessary for magnetic field compatibility, but it exposes

the sample to infrared (IR) radiation from the warmer magnet surroundings. The 4K stage is now only ~1 mm away from the sample holder anchored at the base plate. Although a cryogenic IR filter was installed directly in front of the sample holder to suppress photon inflow via the microwave lines, the dominant contribution to the residual photon population is likely due to direct IR radiation bypassing the copper dowel, as illustrated in Fig.4, and leaking inside the sample holder waveguide. This conclusion is further supported by the temperature of the resonator population, extracted from photon-shot-noise-limited dephasing at the sweet spot. From these measurements, we infer a resonator temperature of approximately 150 mK (see line 196 in the manuscript), which agrees with the measured qubit population corresponding to 165 mK.

We fully agree with the reviewer that minimizing thermal loading is essential for future experiments. To this end, we are exploring improved infrared shielding and thermal anchoring solutions to reduce the effective sample temperature and improve qubit initialization fidelity, while retaining the ability to bias in Tesla fields.

We clarify the difference to Ref.46 in the revised manuscript in Appendix B:

"In contrast, a similar device in the same sample holder geometry was reported in Ref. [46] with a qubit temperature of 37 mK, where improved IR shielding and thermalization was possible due to the absence of the vector magnet."

Furthermore, the fitted spin bath temperature ($T_s = 85$ mK) differs significantly from the qubit temperature. Across cooldowns, T_s ranges from 61 to 298 mK without clear explanation. This discrepancy raises questions about the accuracy of the methodology and suggests the potential for alternative interpretations beyond spin bath freezing due to magnetic polarization.

We thank the reviewer for this insightful comment, which is in line with a comment of reviewer 1 (see above).

We agree that the variation in fitted spin bath temperatures across cooldowns and qubit frequencies is intriguing. However, we caution against overinterpreting these differences, as they arise from fits to extracted flux noise amplitudes rather than direct spectral measurements. In our view, these observations highlight the need for a more detailed spectral characterization of flux noise across field and temperature. Such future studies may help clarify the mechanisms of spin freezing and reveal the microscopic nature of the underlying fluctuators, which promises to provide a deeper insight into the nature of flux noise in superconducting circuits.

To clarify this point in the revised manuscript, we added a sentence in App. D:

"The observed variation in extracted spin bath temperatures motivates further noise characterization, including measurements at various cryostat temperatures and a comprehensive spectral analysis."

Technical Questions and Comments for Improvement

1. **Abstract** – "Suppression of fast flux noise": The term "fast flux noise" is vague. Since noise at different frequencies may have different origins, the specific frequency range under investigation should be clearly stated.

We thank the reviewer for highlighting the ambiguity of the term "fast flux noise". In response, we have updated the abstract to clarify that the suppression we observe pertains specifically to flux noise in the megahertz (MHz) range.

"We also observe a suppression of MHz range fast flux noise in magnetic field, suggesting the freezing of surface spins."

2. **Line 21–27** – Some references (12–19) involve spin qubits coupled to superconducting resonators rather than qubits. These are mainly relevant to spin qubit readout and less connected to the topic of flux noise in superconducting qubits. Consider replacing them with more relevant works.

We thank the reviewer for highlighting the inconsistency. The cited works (Refs. 12–19) indeed focus primarily on spin-resonator interfaces rather than direct qubit-spin coupling. This reflects the practical limitation that superconducting qubits have traditionally been incompatible with the high magnetic fields required for spin systems—a challenge we overcome with the field-resilient qalmonium qubit.

To clarify this distinction, we have revised the manuscript to refer to "superconducting circuits" rather than "qubits" when discussing prior hybrid architectures. The updated sentence now reads:

"Hybrid quantum architectures, where superconducting circuits couple to less amenable but longer-lived, magnetic-field-sensitive DOFs, have already demonstrated impressive achievements, such as coherent spin-photon interactions [12–14], spin ensemble [15–17] and even single-spin detection [18, 19] using superconducting resonators, as well as single-magnon detection with a superconducting qubit [11]."

The revised introduction now better reflects the twofold motivation for field-resilient superconducting qubits: both as sensitive probes of magnetic-field-susceptible DOFs in the qubit environment, and as a platform for advancing hybrid quantum architectures.

3. **Literature Context** – The manuscript would benefit from a broader contextualization of prior flux noise studies in superconducting qubits. Notable references include:

o Quintana et al., *Phys. Rev. Lett.* **118**, 057702 (2017): Noise spectrum over a broad frequency range.

o Lanting et al., *Phys. Rev. B* **79**, 060509(R) (2009): Geometric dependence of low-frequency flux noise.

o Gao et al., *Nat. Commun.* **16**, 3620 (2025): Material dependence of flux noise.

These references reinforce the significance of identifying spin noise origins and would enhance the motivation for the high-field measurements.

We thank the reviewer for pointing out these references. Indeed, they complete the picture of the prior work and they are now included in the revised manuscript:

"We model the flux noise as the sum of a large number of magnetic two-level fluctuators, consistent with the commonly accepted spin-based origin of flux noise [56–58]."

4. **Figure 1f** – Clarify how qubit parameters at different magnetic fields were extracted. Were qubit spectra measured as a function of flux bias at each temperature? How are L_q and L_r independently determined?

The reviewer is indeed correct: the qubit spectrum was measured as a function of flux bias at each magnetic field B_{\parallel} . The spectroscopy was then fit to the fluxonium Hamiltonian (Eq.1) at every value of B_{\parallel} to extract the qubit parameters.

Following the reviewer's remark, we clarify this methodology in the revised manuscript:

"To assess the effect of magnetic field on the fluxonium parameters, we measure the qubit ground to excited transitions frequency f_{ge} near the half- and zero-flux sweet spots at each $B_{||}$, using two-tone spectroscopy (similar to Fig. 1c). We fit f_{ge} to Eq. (1) using the field independent capacitance $C = 1.37$ fF obtained in zero field. From the fitted parameters, using $EJ \propto \Delta(B_{||})$ and $L_{kin} \propto 1/\Delta(B_{||})$, we extract the magnetic field suppression of the superconducting gap in the superinductor ($\delta\Delta L_q$) and nanojunction ($\delta\Delta EJ$), as shown in Fig. 1 f. The suppression of the resonator superconducting gap ($\delta\Delta L_r$) is independently obtained by fitting the resonance frequency shift $f_r \propto 1/\sqrt{L_{kin}}$."

5. Lines 153–158 – The observed Ramsey fringe beating is attributed to critical current fluctuations. Abdurakhimov *et al.*, *PRX Quantum* **3**, 040332 (2022), report a type of TLS affecting critical currents—distinct from standard charge TLSs—that may explain these observations. Please discuss this possibility.

We thank the reviewer for bringing the work of Abdurakhimov *et al.* (*PRX Quantum* **3**, 040332 (2022)) to our attention, which reports a class of "critical-current TLSs": high-frequency (GHz) two-level systems that couple nonlinearly to the qubit via critical current fluctuations, distinct from conventional charge-TLSs. This mechanism is indeed a plausible candidate for the discrete critical current fluctuations we observe in our grAl nanojunction. We now include this hypothesis in the revised manuscript as a motivation for future experiments:

"Further experiments, such as electric field bias or mechanical strain on the substrate [32, 54] or spin-locking TLS spectroscopy [55], are required to identify their cause."

6. Lines 234–235 – The statement that long TLS lifetimes imply a spin environment is questionable. Charge-based TLSs can also exhibit long lifetimes depending on tunneling amplitude.

We thank the reviewer for pointing out that long TLS lifetimes do not directly imply a spin environment, consistent with our observations that the long-lived TLS are not magnetic field susceptible. However the long lifetime makes a spin environment plausible - a hypothesis which is also encouraged by the spin signatures detected with our device. We tone down the statement on the spin-based origin for the long-lived TLS signatures in the revised manuscript:

"These TLSs have been shown to induce non-Markovian qubit dynamics and their long lifetime, exceeding $1/\Gamma_{TLS} \geq 50$ ms, makes a spin-based origin plausible."

7. Line 250 – The cross-relaxation rates used to model TLS hyperpolarization should be clearly described. Given that this technique is relatively new, please include the theoretical model and fitting procedure to make the manuscript self-contained.

Thank you for the remark. We agree with the reviewer that an additional description of the hyperpolarization experiment would be beneficial to improve the clarity of the manuscript. This is also in line with a comment of reviewer 1. Following these remarks, we have added Appendix E and the newly added Figure 9 in the revised manuscript.

8. Lines 631–634 – The high thermal population is attributed to missing thermal shielding. However, this contradicts the claim that the setup is identical to Ref. 46, which reported significantly lower effective temperatures. Please clarify this discrepancy.

We thank the reviewer for highlighting this important inconsistency, which also relates to their earlier comment. As discussed in our response to the first comment, the elevated qubit

temperature arises from infrared radiation in the unshielded vector magnet environment. For clarity, we summarize the relevant explanation below:

In short, the discrepancy between the qubit temperature inferred from thermal population (~ 165 mK) and the cryostat base temperature (~ 30 mK), as well as the lower temperature (~ 37 mK) reported in Ref. 46 can be explained by the different sample-holder environments. In contrast to the setup used in Ref. 46, our sample was now positioned within a 2D vector magnet anchored to the 4 K stage of the cryostat, without additional thermal shielding. This open configuration is necessary for magnetic field compatibility, but it exposes the sample to infrared (IR) radiation from the warmer magnet surroundings. The 4K stage is now only ~ 1 mm away from the sample holder anchored at the base plate. Although a cryogenic IR filter was installed directly in front of the sample holder to suppress photon inflow via the microwave lines, the dominant contribution to the residual photon population is likely due to direct IR radiation bypassing the copper dowel, as illustrated in Fig.4, and leaking inside the sample holder waveguide. This conclusion is further supported by the temperature of the resonator population, extracted from photon-shot-noise-limited dephasing at the sweet spot. From these measurements, we infer a resonator temperature of approximately 150 mK (see line 196 in the manuscript), which agrees with the measured qubit population corresponding to 165 mK.

We clarify the difference to Ref.46 in the revised manuscript in Appendix B:

“In contrast, a similar device in the same sample holder geometry was reported in Ref. [46] with a qubit temperature of 37 mK, where improved IR shielding and thermalization was possible due to the absence of the vector magnet.”